# Insights into the Evolving Epidemiology of *Clostridioides difficile* Infection and Treatment: A Global Perspective

**DOI:** 10.3390/antibiotics12071141

**Published:** 2023-07-01

**Authors:** Crystal Liu, Tanya Monaghan, Abbas Yadegar, Thomas Louie, Dina Kao

**Affiliations:** 1Department of Medicine, University of Alberta, Edmonton, AB T6G 2R3, Canada; crystal.liu@ualberta.ca; 2National Institute for Health Research, Nottingham Biomedical Research Centre, Nottingham Digestive Diseases Centre, School of Medicine, University of Nottingham, Nottingham NG7 2UH, UK; tanya.monaghan@nottingham.ac.uk; 3Foodborne and Waterborne Diseases Research Center, Research Institute for Gastroenterology and Liver Diseases, Shahid Beheshti University of Medical Sciences, Tehran 1985717411, Iran; babak_y1983@yahoo.com; 4Medicine and Microbiology, School of Medicine, University of Calgary, Calgary, AB T2N 1N4, Canada; thomas.louie@albertahealthservices.ca; 5Division of Gastroenterology, University of Alberta, Edmonton, AB T6G 2P8, Canada

**Keywords:** *C. difficile*, epidemiology, CDI trends, ribotypes, evolving treatment, global spread

## Abstract

*Clostridioides difficile* remains an important public health threat, globally. Since the emergence of the hypervirulent strain, ribotype 027, new strains have been reported to cause *C. difficile* infection (CDI) with poor health outcomes, including ribotypes 014/020, 017, 056, 106, and 078/126. These strains differ in their geographic distribution, genetic makeup, virulence factors, and antimicrobial susceptibility profiles, which can affect their ability to cause disease and respond to treatment. As such, understanding *C. difficile* epidemiology is increasingly important to allow for effective prevention measures. Despite the heightened epidemiological surveillance of *C. difficile* over the past two decades, it remains challenging to accurately estimate the burden and international epidemiological trends given the lack of concerted global effort for surveillance, especially in low- and middle-income countries. This review summarizes the changing epidemiology of *C. difficile* based on available data within the last decade, highlights the pertinent ribotypes from a global perspective, and discusses evolving treatments for CDI.

## 1. Introduction

*Clostridioides difficile* is a Gram-positive, anaerobic, spore-forming bacterium that remains an important public health threat globally [1]. *C. difficile* infection (CDI) accounts for approximately 15–20% of all cases of antibiotic-associated diarrhea, ranging from mild diarrhea to pseudomembranous colitis (PMC) [2]. The epidemiology of CDI has been evolving over the past two decades. Most cases of CDI were previously linked to healthcare exposure; however, recent studies have suggested an increased incidence in community-acquired (CA)-CDI reaching up to 40% of all CDI cases [3]. Interestingly, the incidence of multiple recurrent CDI (rCDI) has risen disproportionately to the incidence of CDI. Between 2001 and 2012, CDI increased by 46% while rCDI rose by 189% in the USA [4]. In 2020, the overall incidence rate of CDI in the United States was 101.3 cases per 100,000 persons, with an associated healthcare cost of $6.3 billion USD [5]. With the rising incidence of rCDI, microbial-based therapeutics such as FMT are emerging and being recommended in various treatment guidelines. Furthermore, the Food and Drug Administration has recently approved two stool donor-based products, RBX2660 (Rebyiota) and SER-109 (Vowst), in the treatment of rCDI [6]. Currently, there has been a shift towards microbial restoration as a strategy to combat dysbiosis, the root cause of CDI; however, it remains challenging to accurately estimate the global healthcare burden and costs associated with CDI since epidemiologic surveillance is limited in low- and middle-income countries.

The emergence of a more virulent strain of *C. difficile* has been recognized as one of the principal drivers of the ongoing CDI epidemic. This strain is known as the polymerase chain reaction (PCR) ribotype 027, North American pulse-field type 1 (NAP1), restriction endonuclease analysis (REA) type B1 strain, or simply “ribotype 027”. Ribotype 027 is associated with more severe illness which is refractory to antibiotic therapy and has a greater risk of relapse [7,8,9]. The emergence of this previously uncommon and hypervirulent *C. difficile* strain prompted healthcare facilities to track the incidence and undertake further epidemiological research on *C. difficile.* Over the last decade, new strains have been reported to cause CDI with worse health outcomes, such as ribotypes 014/020, 017, 056, 106, and 078/126 [10,11]. These strains differ in their genetic makeup, virulence factors, and antimicrobial susceptibility profiles, which can affect their ability to cause disease and respond to treatment. As such, understanding *C. difficile* epidemiology is increasingly important to allow for effective prevention measures. This review aims to provide an up-to-date overview of the epidemiology of *C. difficile* globally. It summarizes the changing epidemiology of *C. difficile* based on available data within the last decade, highlights the pertinent ribotypes around the world, and discusses evolving treatments for CDI.

## 2. The Emergence of NAP1/Ribotype 027

The epidemic ribotype 027 has not only been linked to an overall increase in incidence of CDI, but also to increased disease severity and recurrence with higher mortality rates. Starting in the early 2000s, outbreaks of CDI in North America and Europe were associated with ribotype 027 [12]. This strain was a significant predictor of severe CDI and mortality [7,8,9,13,14]. In Canada, from 2009 to 2015, ribotype 027 resulted in higher overall death rates compared to non-NAP1 strains (19.3% vs. 12.3%) as well as higher mortality attributable to CDI (12.2% vs. 1.4%) [13]. The increased disease severity associated with ribotype 027 is thought to be related to mutations in the toxin regulatory gene *tcdC*, a single base pair (bp) deletion at position 117, which leads to a markedly truncated repressor protein, and further on to an 18 bp deletion at position 330–347, which is associated with the more rapid and increased production of toxins A and B [7,15,16,17,18,19,20,21]. In comparison to non-epidemic strains, the genome of the ribotype 027 epidemic strain exhibits the presence of five extra-genetic regions. These additional regions consist of a novel phage island, a two-component regulatory system, and transcriptional regulators [7,15,16,17,18,19]. The increased production of spores has also been suggested as an explanation for the successful spread of this ribotype in the healthcare setting [22]. Following the marked epidemiologic change in CDI with ribotype 027, there have been a significant number of surveillance studies that have investigated the emergence of new epidemic strains, given that new strains continue to emerge.

Since then, molecular studies have shown two distinct lineages of ribotype 027, labelled FQR1 and FQR2, which have both acquired a fluoroquinolone-resistant mutation in parallel [16]. This is due to significant selective pressure, with fluoroquinolone antibiotics being one of the most commonly prescribed antibiotic classes in the late 20th and early 21st centuries [23]. The FQR1 lineage has spread throughout the world, most likely originating in the United States and spreading to Asia and Switzerland, whereas the FQR2 lineage was associated with a more widespread distribution initially, with rapid population expansion in the United States, Canada, and more widely throughout Europe [16]. The emergence of this previously uncommon and hypervirulent *C. difficile* strain prompted healthcare facilities to track the incidence and undertake further epidemiological research on *C. difficile.*

While these clonal strains are associated with increased disease severity, it is now recognized that the strain type does not always correlate with disease severity, especially in non-epidemic settings [24,25]. The hypervirulence of certain strains has recently become a topic of debate. In fact, recent cross-sectional and single-center studies have shown no association in disease severity between ribotype 027 and 078 [26,27]. Given the complexity of the host immune response that determines disease severity, further investigations are needed to better understand the connection between strain type and disease severity.

## 3. *C. difficile* Epidemiology in North America

In North America, the incidence and severity of CDI have been well-documented. In the United States, the most recent 2020 surveillance data by the Center for Disease Control and Prevention showed an incidence of 101.3 cases per 100,000 persons, with 51.2% being CA-CDI and 50.1% being healthcare associated (HA)-CDI [28]. The most common CA and HA ribotypes in the United States over the past decade were ribotypes 027, 106, 014/020, and 002 [29,30]. Notably, ribotype 027 decreased significantly among both HA and CA isolates in the United States [30]. Other virulent strains associated with more severe disease, including ribotypes 078 and 244, are low in prevalence in the United States [11,30,31]. Within the last decade, the estimated burden of CDI in the United States has decreased, despite the increasing use of more sensitive nucleic acid amplification tests (NAAT) [30].

The incidence of CDI, particularly HA-CDI, appears to have reached a plateau in recent years after increasing steadily over the previous decade [32]. Over the past decade, HA-CDI decreased by an average of 6% annually [29]. Many studies have described the association between fluoroquinolone use and CDIs [7,33,34]. The reduction in fluoroquinolone use is associated with lower HA-CDI, with decreasing prevalence of epidemic strains, such as ribotype 027, likely due to decreased selective pressures [30,35,36]. The most recent survey shows that the prevalence of ribotype 027 is approximately 10% [37]. The decline in HA-CDI and ribotype 027 followed a series of infection prevention programs, and the institution of hand hygiene, contact precautions, disinfection practices, and financial reimbursement penalties [30,36]. This further emphasizes the importance of utilizing modifiable risk factors to minimize the occurrence of resistant *C. difficile* strains.

Similarly, in a Canada-wide CDI study from 2009 to 2015 by the Canadian Nosocomial Infection Surveillance Program, the national rate of HA-CDI decreased from 5.9 to 4.3 per 10,000 patient days [38]. Over the past decade, while ribotype 027 remained the predominant strain, there was a substantial decrease in its prevalence from 25% to 9.4%, along with an increase in the proportions of other strains such as ribotype 106 and 014/020 [13]. Currently, the most prevalent CA-CDI is ribotype 106 [30]. From 2015 to 2019, the incidence of ribotype 106 has been increasing from 7.3% to 18.1% [13]. Ribotype 106 was initially identified in the United Kingdom in 1999, and it is now one of the most predominant strains in North America [13,39]. Furthermore, this ribotype is associated with a greater likelihood of rCDI [39,40]. It has an enhanced ability to produce spores, form biofilms, and persist in hospital settings.

In recent North American surveillance studies, the antimicrobial resistance rate of *C. difficile* for vancomycin ranges from 1.2 to 2.1%, with some studies suggesting an increase in vancomycin resistance over the past decade [41,42,43]. All isolates were metronidazole susceptible, with a resistance rate of 1.9–2.7% [41,42,43]. Fidaxomicin demonstrated the lowest MIC of all antimicrobial agents at an MIC_90_ of 0.5 mg/L [41]. Despite the decline in the prevalence of ribotype 027, its isolates continue to show increased resistance to most antimicrobials, including a two-fold higher MIC compared to other isolates [41]. Among rare fidaxomicin-resistant isolates, mutations in the *rpoB* gene have been postulated to interfere with the fidaxomicin binding site [44,45]. Fortunately, in an in vivo animal model, the same *rpoB*-positive isolates have been demonstrated to exhibit lower cytotoxicity, attenuated growth, poor sporulation, and decreased toxin A/B generation [44,45]. Overall, there is heterogeneity between studies with respect to the reported resistance rates. However, interpreting resistance values which utilize systemic concentration breakpoints in the treatment of luminal infections, where concentrations are exceedingly high, is a challenge.

## 4. *C. difficile* Epidemiology in Europe

In a 2022 Europe-wide survey on the incidence of CDI involving 559 hospitals performed by the European Centre for Disease Prevention and Control (ECDC), the mean incidence of CDI was 3.48 cases per 10,000 patient days [46]. The majority of cases (60.9%) were HA-CDI, while 32.7% accounted for CA-CDI cases [46]. In general, ribotypes have become more heterogenous, given a significant decrease in the dominant ribotype 027. The most common PCR ribotypes were ribotype 014 (16.8%) and ribotype 078 (7–11% of cases) [11,46,47]. Specifically, ribotype 078 was commonly detected in regions and jurisdictions where there is a high prevalence of pigs, such as in Belgium, Czechia, Ireland, and the Netherlands [46]. Ribotype 014 and 078 are livestock-associated (LA) strains, although these ribotypes have recently emerged as pathogens that cause increased virulence and disease severity in humans [48]. Specifically, ribotype 078 is also more likely to be CA strain and affect younger individuals [11]. Genome comparison studies revealed that ribotype 078, along with other clade 5 strains, is divergent from the rest of the species, separating approximately 1.1–85 million years ago [48]. The current understanding is that ribotype 078 emerged from a non-toxigenic ancestor strain that has horizontally acquired a pathogenic locus (PaLoc), which provided it with its virulent properties [48].

Previous reports of *C. difficile* in food, along with the link between ribotypes 078 and 014/020 with pig farming, suggests that the food chain may be a vector for these ribotypes [11,49,50,51]. A European study noted that some *C. difficile* isolates have shown a distinct pattern of genetic relationship that does not cluster within a hospital, region, or country. This suggests that the dissemination of these *C. difficile* strains is not through person-to-person transmission, but rather, transmission through other channels, such as the food chain [30,50,51]. Although there is currently no concrete evidence supporting the transmission of *C. difficile* from animals to humans, there is a growing suggestion that it should be regarded as a zoonotic pathogen. Ongoing endeavors are being made to enhance our comprehension of *C. difficile* transmission in this setting.

The hypervirulent ribotype 027 was the third most frequently reported ribotype in Europe, with its relative prevalence decreasing compared to prior years. The highest prevalence of ribotype 027 was found in Hungary (67.6%), Poland (63.0%), and Slovenia (44.4%), and relatively lower proportions in all other countries (2.5%) [46]. In England and the Netherlands, the implementation of a national ribotyping service was associated with the control of the hypervirulent ribotype 027 strain and coincided with a marked reduction in CDI incidence and related mortality [52]. Further, similarly to North America, the restriction of fluoroquinolone prescriptions is thought to have contributed to a reduction in CDI due to the decreased selection pressure of fluoroquinolone-resistant strains, including ribotype 027 [53].

Central Europe reported high proportions (90.7%) of ribotypes similar to that of 027 within clade 2, which include ribotype 036, 198, 176, and 181 [46]. Of concern, there are high proportions of metronidazole resistance within ribotype 027 and ribotype 027-like strains [46]. In a recent European longitudinal surveillance on *C. difficile* antimicrobial resistance, the mean MIC of metronidazole and vancomycin were 0.46 mg/L and 0.70 mg/L, respectively, with reduced metronidazole susceptibility seen in ribotype 027 and 198 [54].

## 5. *C. difficile* Epidemiology in the Rest of the World

While the incidence of HA-CDI has plateaued or declined in North America and Europe, it seems to be rising in Australia. In 2013, the average yearly rate of CDI was 3.94 per 10,000 patient bed days and this increased to 4.05 per 10,000 patient days in 2018. Australia also has a markedly different pool of *C. difficile* strains compared to other parts of the world [55]. A recent Australian longitudinal surveillance, the *C. difficile* Antimicrobial Resistance Surveillance (CDARS) study, showed that ribotype 014/020 (29.5%) was the most prevalent strain and ribotype 126 was the most prevalent toxin-positive strain [56]. Ribotype 027 continued to be infrequent in Australia [56]. The porcine-associated ribotype 014/020 is thought to be non-hypervirulent, while ribotype 078/126 is associated with increased severity and poor disease outcomes [51,57]. Recent surveillance studies within Australia have shown good susceptibility of all strains to metronidazole, fidaxomicin, rifaximin, and amoxicillin-clavulanate, with a low resistance rate to meropenem, moxifloxacin, and vancomycin [58]. Overall, the antimicrobial resistance of *C. difficile* strains in Australia remains low.

In Asia, there are limited data on the prevalence of CDI; however, a recent metanalysis showed a pooled estimated 5.3 episodes of CDI per 10,000 patient days [59]. Overall, there appears to be a higher prevalence of CDI in East Asia compared to South Asia and the Middle East [59]. Ribotyping data have identified a unique set of toxicogenic ribotypes, including 001, 002, 010, 014, 017, 018, and 046 [60,61,62,63]. Ribotype 027 and 078, both of which are prevalent in North America and Europe, are noted to be infrequently detected in Asia [11,59,63]. In the Middle East, a recent large-scale cross-sectional study has revealed that ribotypes 001, 126, and 084 were the most frequent ribotypes among CDI patients from healthcare settings [64]. There is less data on the molecular epidemiology of CDI in South America, although ribotypes 027, 106, 012, 046, and 014/020 are the most common strains [65].

Ribotype 017, an A-B+ strain that is widespread in Asia, has been noted to be the dominant ribotype in India, Thailand, Indonesia, parts of South Africa, along with up to 48% of isolates in China [66,67,68,69,70,71,72]. Ribotype 017 is thought to have originated in Asia and spread globally. Since then, ribotype 017 has been responsible for multiple outbreaks internationally and is now emerging as a significant pathogenic strain of CDI [73]. The significance of the A-B+ *C. difficile* strains have been previously under-recognized, given the prior assumption that both TcdA and TcdB were required for a virulent phenotype [74]. After the association between *C. difficile* and PMC was shown, the initial belief was that TcdA was required to cause initial damage to the intestinal mucosa before TcdB could exert its cytotoxic effect [75]. As a result, until the early 2000s, there was a shift towards using diagnostic studies involving rapid immunoassays for only the detection of TcdA, leading to likely underestimation of A-B+ strains [49,76,77]. The current understanding is that TcdB can exert its action in the absence of TcdA, and the human intestinal mucosa is 10 times more sensitive to TcdB than TcdA [73]. This emphasizes the importance of using standardized *C. difficile* diagnostic methods that include both toxins. Recent studies have shown that it is possible to ribotype *C. difficile* directly from fecal DNA, with sensitivity and specificity comparable to diagnostic toxin gene qPCR and conventional DNA typing [78,79].

Despite the disease burden of *C. difficile* in low- and middle-income countries, epidemiological data assessing the burden of CDI remain relatively scarce [80,81]. Most epidemiological studies on CDI have been reported from North America and Europe over the past two decades, with few reports from Latin and South America, Africa, and Asia [81]. A significant effort to enhance CDI awareness and improve laboratory capacity, along with a coordinated national effort to strengthen epidemiological tracking of CDI, is necessary in these regions. A summary of evolving CDI epidemiology and ribotypes around the globe can be found in Table 1 and Table 2.

## 6. Evolving CDI Treatments

Currently, there are no specific recommendations regarding treatment in accordance with different strain characteristics. Treatments are instead tailored to the severity of the clinical presentation. Metronidazole used to be the first line therapy for CDI. Due to decreasing clinical response rates observed since the early 2000s, metronidazole is no longer recommended as the first line therapy, although it is still used in clinical practice, especially in individuals less than 65 years of age with a mild initial episode of CDI [82]. Clinical practice guidelines from the Infectious Diseases of America (IDSA) and Society for Healthcare Epidemiology of America (SHEA) recommended treating the initial CDI episode with either vancomycin or fidaxomicin, with both options being preferable to metronidazole in 2017 [83]. However, newer therapeutic options, including fidaxomicin, bezlotoxumab, fecal microbiota transplantation, and other live biotherapeutics have emerged and are briefly reviewed below.

### 6.1. Fidaxomicin

Compared with vancomycin treatment, treatment of CDI with fidaxomicin is associated with a ~50% lower rate of recurrence of CDI in non-NAP1 strains. Fidaxomicin, a bactericidal antibiotic active against Gram-positive anaerobes, has been available and approved by the FDA since 2011. Fidaxomicin is thought to have a lower impact on the normal gut microbiota than oral vancomycin.

Three double-blind RCTs and one open label RCT have compared fidaxomicin and the standard vancomycin regimens to date, with the pooled results showing a sustained response of CDI to fidaxomicin compared to standard vancomycin, with comparable initial clinical cure [84,85,86,87]. This makes fidaxomicin a less costly treatment strategy compared to vancomycin for the treatment of CDI in older patients [88], although further cost-effective studies are needed in this area [89]. Of note, the same reduction in recurrence compared to vancomycin is not demonstrated in patients infected with ribotype 027 [12,15]. In 2021, the IDSA and SHEA recommended the use of fidaxomicin rather than the standard course of vancomycin as the preferred therapy for the initial episode of non-fulminant CDI [89]. There are limited data regarding the use of fidaxomicin in fulminant CDI cases since these patients were excluded from the clinical trials [85]. Despite the clinical benefits, uptake and implementation remain slow given resource limitations and high costs. We anticipate that the uptake of fidaxomicin would increase with reduced cost and improved accessibility.

### 6.2. Antibody-Mediated Therapy

In addition to antibiotics, the use of monoclonal antibodies has emerged as a novel adjunctive therapy for the treatment and prevention of CDI recurrence. In 2016, the FDA approved bezlotoxumab, a monoclonal antibody that binds to *C. difficile* toxin B, for the treatment of patients with a high risk of CDI recurrence. Two double-blind RCTs found that the rate of recurrence of CDI with bezlotoxumab alone was significantly lower than with the placebo, with similar rates of adverse reactions compared with the placebo [90]. It also shows efficacy against the hypervirulent ribotype 027 [91]. A potential safety signal was the occurrence of heart failure with bezlotoxumab, although the risk is low [90,92]. Currently, the conditional recommendation from the IDSA and SHEA is for co-intervention with bezlotoxumab along with CDI-directed antibiotics in patients with rCDI episodes within the past 6 months [89]. The use of bezlotoxumab is, however, limited by the high cost and logistics.

In contrast to monoclonal antibody therapy, polyclonal antibody therapy is currently still in development. Intravenous immunoglobulin (IVIG), with its use in numerous other clinical conditions, has also been proposed for use in CDI. It has only been examined in retrospective studies, often in the context of severe rCDI [93,94,95]. There are promising studies that show specific binding and neutralizing antibodies to *C. difficile* antigens in select IVIG preparations [96]. However, randomized studies are needed to demonstrate a therapeutic effect.

### 6.3. Fecal Microbiota Transplant (FMT)

During the last decade, FMT has emerged as an effective treatment to prevent CDI recurrence. FMT involves the infusion of donor stool into the gastrointestinal tract of recipients. The primary indication for FMT is three or more episodes of mild to moderate CDI, or at least two episodes of CDI resulting in hospitalization or significant morbidity [97]. Several studies have shown that FMT is superior to oral vancomycin and fidaxomicin in the setting of rCDI [98,99]. Recurrent CDI can occur in 20–25% of patients, with increasing rates following each subsequent episode, up to 40–45% in patients after the second CDI episode, and more than 65% after three or more CDI episodes [100,101,102]. A subset of patients experience chronic relapsing CDI with multiple recurrences which is an ongoing challenge [102]. From 2001 to 2012, there was a 189% increase in the annual incidence of rCDI [4]. FMT has shown remarkable efficacy and safety in the treatment of rCDI. The clinical efficacy after the initial FMT was 84% and rose to 91% after a repeat FMT in a pivotal randomized controlled trial [103]. Early FMT was associated with reduced mortality rate in an outbreak of ribotype 027 infections [104].

Despite these successes, there remain some concerns regarding the short-term safety of FMT due to reported cases of bacteremia and death associated with donor-derived pathogens [105,106]. Furthermore, the long-term follow-up registry study is still ongoing and is required to further inform the safety profile data [107]. Additionally, while there may be some differences in the clinical efficacy based on how FMT is delivered, whether by enema, colonoscopy, nasogastric tube, or oral capsules, there are benefits and risks associated with each route of delivery. Ultimately, the best option may have to be chosen based on product availability, patient factors, and practitioner expertise.

### 6.4. Emerging Therapies

Due to the potential risks of FMT, there is a significant interest in the development of safer, donor-independent defined microbial consortia for the modulation of gut microbiota in CDI. VE-303 is a bacterial consortium comprised of eight strains of commensal Clostridia in adults at high risk of rCDI. A recent phase II randomized control trial with 79 subjects comparing high-dose VE303 with a placebo found that VE303 significantly reduced the risk of rCDI by 32% relative to the placebo [108]. Another preparation, SER-109, is an oral formulation developed by Seres Therapeutics that encompasses approximately 50 species of Firmicutes spores derived from human stools, with subsequent inactivation of potential pathogens, and recently received FDA approval in April 2023. Their phase III randomized control trial with 182 participants had shown SER-109 to be superior to the placebo in reducing the risk of rCDI by 28%, with a good safety profile [109]. Microbial Ecosystem Therapeutic 2 (MET-2) is another oral formulation that consists of 40 lyophilised bacterial species that were originally derived from a healthy donor, then synthesized independently to eliminate the potential risks of donor-derived pathogens. MET-2 had been evaluated in a phase I trial, which found MET-2 treatment prevented recurrent infection in 79% of adults 40 days after the initial treatment and had been safe and well-tolerated among individuals with rCDI [110]. In the next decade, the infusion of select microbial consortia represents an innovative treatment in patients with CDI. Given the cost implication of these novel microbiota-based therapies, further price reduction would be necessary before they can be implemented widely.

Emerging therapeutic targets to treat CDI are in development. Phage therapy is an experimental treatment for CDI which involves the use of bacteriophages; viruses selected to target and kill *C. difficile* bacteria. In preclinical studies, phage-based therapy has shown promise in inhibiting spore outgrowth in vitro [111,112,113,114,115,116,117,118]. For instance, Mondal et al. identified that recombinantly expressed cell wall hydrolase lysin from *C. difficile* phage phiMMPo1 was active against *C. difficile* [113]. The potential advantage of phage therapy is the ability to specifically target *C. difficile* bacteria without harming the beneficial bacteria in the gut microbiome. This could potentially reduce the risk of rCDI. The challenges with phage therapy include its stability in the gastrointestinal tract, the need to select and match phages to specific *C. difficile* strains, potential for phage resistance over time, and the need for ongoing research to determine the optimal administration strategies [111,112,113,114,115,116,117,118]. There are further promising pre-clinical data on emerging therapies in clinical trials which have been reviewed elsewhere [117].

## 7. Conclusions

*C. difficile* remains a significant cause of HA infections and is increasingly being recognized as a CA pathogen, with an increased incidence of rCDI in recent years. It is important to understand *C. difficile* epidemiology in order to achieve ongoing coordinated surveillance programs to aid in identifying cases, monitor trends, and detect potential reservoirs. Implementation of standardized laboratory testing is also critical to accurately characterize the nature of CDIs, especially in low- and middle-income countries. Further, preventing CDIs involves ongoing antimicrobial stewardship, because even effective medications such as vancomycin and fidaxomicin may lead to the development of resistance down the line. The pathogenesis of CDI is characterized by microbial dysbiosis; hence, the field is progressing towards microbial-based therapeutics such as FMT and donor-independent microbial consortia. Ideally, as the potential risks associated with microbial therapies decrease, along with increased cost-effectiveness, these treatments will be integrated earlier in the treatment paradigm.antibiotics-12-01141-t002_Table 2Table 2A global comparison of prevalent ribotypes of *C. difficile* in different regions of the world.Nation/RegionPrevalent StrainsReferenceUnited StatesRibotypes 027, 106, 014/020, 002, 001CDC 2020 [28], Lessa et al. [119], Kim et al. [29], Guh et al. [30]CanadaRibotypes 027, 106, 014/020Katz et al. [13], Du et al. [13], Carlson et al. [39]EuropeRibotypes 014/020, 078, 027, 001ECDPC 2022 [46], Freeman et al. [47]AustraliaRibotypes 014/020, 126, 078/126Hong et al. [56], Putsathit et al. [58]AsiaRibotypes 017, 018, 014/020, 001, 002, 010, 046, 126, 084Collins et al. [81]South AmericaRibotypes 027, 106, 012, 046, 014/020Diniz et al. [64], Salazar et al. [120]

## Figures and Tables

**Table 1 antibiotics-12-01141-t001:** A global comparison of CDI burden and epidemiology in different regions of the world.

Nation/Region	Incidence of CDI	Reference
United States	Incidence of 101.3 [CA (51.2) and HA (50.1)] cases per 100,000 persons in 2020; incidence of 148.55 [CA (65.81) and HA (82.74)] cases per 100,000 persons in 2015	CDC 2020 [28]
Canada	National rate of HA-CDI decreased from 5.9 to 4.3 per 10,000 patient days from 2009 to 2015	Katz et al. [38]
Europe	Mean incidence of CDI was 3.48 cases per 10,000 patient days in 2016–2017; 60.9% HA, 32.7% CA, and 6.7 rCDI	ECDC 2022 [46]
Australia	HA-CDI 3.94 per 10,000 patient bed days in 2013 and 4.05 per 10,000 patient bed days in 2018	ACSQHC 2020 [55]
Asia	Pooled incidence rate at 5.3 per 10,000 patient days	Collins et al. [81]

## Data Availability

No new data were created or analyzed in this study. Data sharing is not applicable to this article.

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
