# Peer review of "Insights into the Evolving Epidemiology of Clostridioides difficile Infection and Treatment: A Global Perspective"

_antibiotics, 2023, doi:10.3390/antibiotics12071141_

Round 1

Reviewer 1 Report

This article did an updated review about CDI. Overall, the manuscript is well-written. I just have one minor suggestion.

1. Please add at least two tables or figures in this review article.

Author Response

Two tables have been included to summarize the relative CDI burden and prevalent ribotypes around the world. Thank you!

Reviewer 2 Report

The authors have made a meticulous effort summarizing the epidemiology of Clostridioides difficile epidemiology. I  feel  the authors should add some literature on CDI recurrence too.

Author Response

We have included a section surrounding recurrent C. diff in the introduction and part of the treatment section. Thanks!

Interestingly, the incidence of multiple recurrent CDI (rCDI) has risen disproportionately to the incidence of CDI; between 2001 and 2012, CDI increased by 46% while rCDI rose by 189% in the USA [4]. (Line 36-38)

Recurrent CDI can occur in 20-25% of patients, with increasing rates following each subsequent episode, up to 40-45% in patients after the second CDI episode, and more than 65% after three or more CDI episodes [99-101]. A subset of patients experience chronic relapsing CDI with multiple recurrences which is an ongoing challenge [101]. From 2001 to 2012, there was 189% increase in the annual incidence of mrCDI [4]. (Line 304-310)

Reviewer 3 Report

This manuscript deals with the epidemiology of Clostridioides difficile and the perspective of controlling infections caused by it.

This review is of public health interest as it could help implement protocols and safety measures to limit the spread of Clostridioides difficile infections.

However, the reviewer would like to draw the attention of the authors to a few remarks.

Obviously, reviews of this type should answer several questions.

The first question is: why do this research? It is clear that Clostridioides difficile infection is a serious public health threat in many regions. But there are similar studies, see, for example, the first reviews that came across in the Google Academy

Brestrich, G., Angulo, F.J., Berger, F.K. et al. Epidemiology of Clostridioides difficile Infections in Germany, 2010–2019: A Review from Four Public Databases. Infect Dis Ther 12, 1057–1072 (2023). https://doi.org/10.1007/s40121-023-00785-2;

Younas, M., Royer, J., Weissman, S.B. et al. Burden of community-associated Clostridioides difficile infection in southeastern United States: a population-based study. Infection 48, 129–132 (2020). https://doi.org/10.1007/s15010-019-01368-5;

Finn, E., Andersson, F.L. & Madin-Warburton, M. Burden of Clostridioides difficile infection (CDI) - a systematic review of the epidemiology of primary and recurrent CDI. BMC Infect Dis 21, 456 (2021). https://doi.org/10.1186/s12879-021-06147-y;

Dirks EE, Luković JA, Peltroche-Llacsahuanga H, Herrmann A, Mellmann A, Arvand M. Molecular Epidemiology, Clinical Course, and Implementation of Specific Hygiene Measures in Hospitalized Patients with Clostridioides difficile Infection in Brandenburg, Germany. Microorganisms. 2023; 11(1):44. https://doi.org/10.3390/microorganisms11010044;

And so on. The reviewer believes that authors should confirm that similar reviews do not exist in order to avoid duplication. Perhaps this should indicate the time frame and search conditions that are covered by this overview.

Second question. What can be learned from this study? Since the purpose and conclusions of this review are not clearly described, it is difficult to answer this question. It seems that this can be corrected if, at the end of each section, the authors would express their opinion on the results that they describe.

This also applies to the "Conclusion" section.

References in the text are not formatted according to the rules of the journal, in the manuscript there are round brackets (according to the formatting rules, square brackets are required).

The statement that is in lines 42–45 seems to need to be reformulated, since in its present form the meaning that the authors wanted to convey to the reader is not clear. Probably, here we were talking about naming options for the virulent strain of C. difficile, which is “ribotype 027”?

Round 2

Reviewer 3 Report

The reviewer believes that the authors have sufficiently answered most of the questions and comments.

This manuscript can be accepted for publication after minor adjustments to correct the parentheses in the text for references to square brackets, as required by the rules of the journal.